# Realtime Video Frame Interpolation using One-Step Diffusion Sampling

**Yongrui Ma**[1,2], **Shijie Zhao**[1*], **Mingde Yao**[2,4], **Junlin Li**[1], **Li Zhang**[1], **Xiaohong Liu**[5],
**Qi Dou**[3], **Jinwei Gu**[3], **Tianfan Xue**[2,4*]
[1]ByteDance Inc.    [2]CUHK MMLab    [3]CUHK    [4]CPII under InnoHK    [5]SJTU

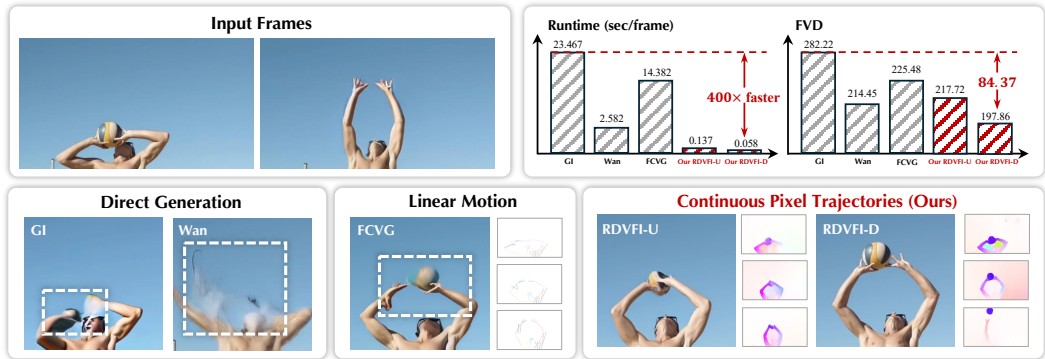

Figure 1: Video frame interpolation results of the proposed Unet-based RDVFI-U and DiT-based RDVFI-D with comparisons to state-of-the-art methods. RDVFI produces the most visually pleasant and the best numeric results with a $400\times$ acceleration, owing to more proper modeling of the intermediate motions using the high-order continuous pixel trajectories, compared with previous direct frame-generation methods, and linear motion controls.

## Abstract

Video Frame Interpolation (VFI) involving large, complex motions remains a significant challenge due to the difficulty of modeling diverse pixel trajectories from limited inputs. Traditional methods struggle with low-order approximations, and recent Latent Video Diffusion Models (LVDM) improve it through a conditional generation modeling. Still, current LVDMs often prioritize pixel fidelity over motion coherence in their reconstruction objective, leading to artifacts in extreme motion scenarios. To address this, we propose RDVFI, a novel approach that leverages an LVDM to generate sparse latent keyframes which define high-order, continuous pixel trajectories. The estimated continuous pixel trajectories accurately index pixel movements from inputs to arbitrary timestamps, generating optical flows to warp input pixels into the target frame. By decoupling sequence motion generation from high-resolution rendering, RDVFI operates on a fixed, lower resolution, and fewer diffusion sampling steps, introducing significant efficiency gains. Extensive experiments demonstrate that RDVFI achieves state-of-the-art visual and numerical performance, with over 75% of viewers selecting it as the best method in terms of motion and frame quality compared to leading baselines. Furthermore, RDVFI is the first LVDM-based VFI method to achieve real-time performance (17 FPS at $1024 \times 576$), offering a $\times 44$ acceleration over the current state-of-the-art and also robustly handling challenging motions. Please visit the project page for code and more visualization samples.

## 1 Introduction

Video Frame Interpolation (VFI) aims to predict intermediate frames between two input ending frames. It serves as a foundational component in diverse fields, including high-frame-rate photogra-

---

*Corresponding authors.

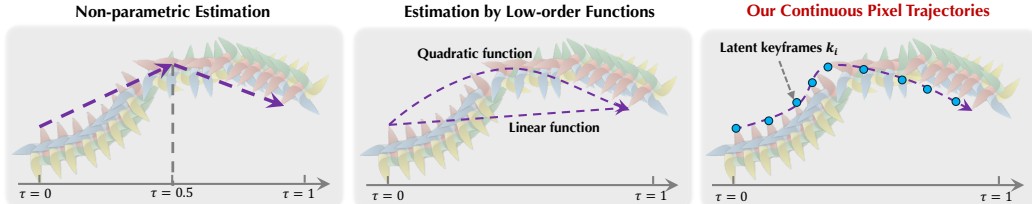

Figure 2: Comparisons between motion estimation methods for VFI. Previous non-parametric methods (Huang et al., 2022; Reda et al., 2022; Li et al., 2023; Danier et al., 2024; Lew et al., 2025) and low-order approximations (linear (Jiang et al., 2018; Bao et al., 2019)/quadratic (Xu et al., 2019; Liu et al., 2020)) struggle to capture large and nonlinear motions like the above rotating windmill. The proposed RDVFI solves continuous pixel trajectories via high-order functions determined by sparse latent keyframes ($\{k_i\}_{i=1}^N$) from the LVDM. Therefore, RDVFI can interpolate to more diverse and complex real-world motions that previous methods fail to capture.

phy (Xiang et al., 2020; Gao et al., 2022; Ma et al., 2024), cinematic post-production (Kokaram et al., 2020; Briedis et al., 2021), and scientific visualization (Priessner et al., 2024; Joshi et al., 2025). Current methods often interpolate by estimating optical flows between intermediate and input frames to warp input pixels to each interpolated frame. As illustrated in Fig. 2, standard non-parametric methods (Huang et al., 2022; Reda et al., 2022; Li et al., 2023; Guo et al., 2024; Danier et al., 2024; Lew et al., 2025) predict flows between one middle and input frames directly in a data-driven manner, optionally assisted by an image diffusion model. While straightforward, these approaches struggle to resolve nonlinear, non-rigid motions. Some methods approximate pixel trajectories using rigid motion priors, such as linear (Jiang et al., 2018; Bao et al., 2019) or quadratic (Xu et al., 2019; Liu et al., 2020) functions. However, these low-order approximations lack the ability to model the complex and ambiguous real-world dynamics.

Recently, Latent Video Diffusion Models (LVDM) have revolutionized generative tasks by their ability to produce high-fidelity video content (Blattmann et al., 2023; Guo et al., 2023; Wang et al., 2025). Inspired by this, state-of-the-art approaches also (Zhu et al., 2024; Wang et al., 2024; 2025) reformulate VFI as a conditional generation problem using LVDMs. Still, despite their ability to synthesize plausible content, these models face two critical limitations. First, they rely on standard pixel-reconstruction objectives that prioritize local texture approximation over global motion coherence. This often leads to severe structural artifacts, such as ghosting and object deformation, as seen in Fig. 1. Second, the iterative diffusion sampling required for high-frequency detail imposes a heavy computational burden. Although current distillation techniques can accelerate these methods by reducing the number of sampling steps, they may aggravate the structural artifacts in interpolated frames, especially when dealing with nonlinear and non-rigid motions.

In this paper, we propose RDVFI, a novel VFI framework that formulates large and complex motions as high-order functions, with coefficients determined by an LVDM. Previous methods model pixel movements based solely on input frames, limiting them to linear (Jiang et al., 2018; Bao et al., 2019), quadratic (Xu et al., 2019; Liu et al., 2020), or nonparametric (Huang et al., 2022; Sim et al., 2021; Li et al., 2023; Danier et al., 2024; Lew et al., 2025) formulations, as shown in the first two examples in Fig. 2. Instead, by incorporating an LVDM to generate additional latent keyframes, RDVFI enables high-order pixel trajectories, which capture real-world dynamics more accurately than previous approaches (see Fig. 2), as shown in the third example of Fig. 2. RDVFI then samples the estimated continuous pixel trajectories for optical flows to warp input pixels for interpolation. Because our LVDM primarily focuses on generating accurate intermediate motions that lack high-frequency details, the proposed RDVFI can interpolate to complex motions with superior temporal stability via one-step diffusion sampling, introducing significant efficiency gains in Fig. 1.

RDVFI offers three primary advantages. First, by shifting the LVDM's objective from frame synthesis to motion modeling, we achieve superior temporal coherence and accuracy, as shown in Fig. 1. Second, because the predicted trajectories represent underlying structural motion rather than fine-grained texture, one-step sampling is sufficient to capture the necessary dynamics, significantly reducing inference latency. Third, as motion can be computed from sparse and low-resolution keyframes, RDVFI further accelerates the inference by offloading major computation on

a low-resolution and sparse input, and uses a relatively lightweight module for dense-frame and full-resolution computation. This design makes RDVFI extremely efficient for high-resolution input.

Extensive evaluations demonstrate that RDVFI successfully interpolates large-scale motions with over $50\times$ acceleration of the leading baseline, Wan (Wang et al., 2025). On a single A100 GPU, RDVFI processes $1024 \times 576$ resolution at 17 FPS. Our core contributions are:

- We present RDVFI, a VFI framework that solves large, complex motions using high-order continuous pixel trajectories and coefficients determined by a one-step LVDM. To our knowledge, RDVFI is the first work to solve motion in VFI using high-order functions or one-step LVDM.
- We introduce an implicit motion regularization term that **enforces physical and temporal consistency**, allowing for accurate interpolation of complex, non-linear trajectories.
- We demonstrate significant gains in both perceptual quality and computational efficiency, effectively bridging the gap between diffusion-based generative power and practical deployment requirements.

## 2 RELATED WORK

### 2.1 VIDEO FRAME INTERPOLATION

**Non-generative Methods** Conventional VFI methods (Jiang et al., 2018; Bao et al., 2019; Xu et al., 2019; Liu et al., 2020; Huang et al., 2022; Reda et al., 2022; Li et al., 2023) primarily rely on regression-based objectives. These are categorized by their underlying motion priors. Early approaches assume pixels follow rigid, predefined trajectories, such as linear (Jiang et al., 2018; Bao et al., 2019) or quadratic (Xu et al., 2019; Liu et al., 2020) models. However, these parametric assumptions fail to capture the stochastic and non-linear nature of real-world dynamics. Subsequent data-driven strategies (Huang et al., 2022; Reda et al., 2022; Li et al., 2023; Guo et al., 2024) predict optical flows between specific timestamps and input frames. While effective at reducing warping artifacts, these methods remain brittle when confronted with the large and complex motions.

**Generative-based Methods** Diffusion Models (DMs) (Ho et al., 2020) have demonstrated superior performance in solving ill-posed inverse problems for image (Dhariwal & Nichol, 2021; Ramesh et al., 2022; Saharia et al., 2022) and video synthesis (Hong et al., 2023; Blattmann et al., 2023; Yang et al., 2024; Lin et al., 2024; Wang et al., 2025). Recent studies (Danier et al., 2024; Lew et al., 2025; Zhu et al., 2024; Wang et al., 2024; 2025) leverage DMs to improve interpolation fidelity. For instance, LDMVFI (Danier et al., 2024) and MoMo (Lew et al., 2025) use image-based DMs to refine flow estimation. However, they remain optimized for relatively simple motion. More recent state-of-the-art frameworks like GI (Wang et al., 2024), FCVG (Zhu et al., 2024), and Wan (Wang et al., 2025) treat VFI as a conditional generation task using Latent Video Diffusion Models (LVDMs). Despite the progress, these methods suffer from a "pixel-fidelity bias": their training objectives focus on per-pixel reconstruction rather than explicit motion consistency and accuracy. Consequently, they often produce frames with rich frame details but poor temporal coherence and unnatural motions.

### 2.2 ENHANCING MOTION IN LVDMs

Several attempts (Zhu et al., 2024; Chefer et al., 2025; Shi et al., 2024) have been made to improve motion modeling within LVDMs. FCVG (Zhu et al., 2024) utilizes linearly interpolated sparse matching to guide motions, yet it remains limited by the reliability of the matching algorithm and the simplicity of its motion model. VideoJAM (Chefer et al., 2025) forces LVDMs to jointly predict frames and flow visualizations; however, this multitask approach often leads to frame degradation due to a distribution mismatch between natural imagery and flow visualizations. Alternatively, Motion-I2V (Shi et al., 2024) decouples flow from frame generation using separate LVDMs. However, it still fails on complex motions because flow estimation degrades with increasing temporal distance, making it impossible to generate accurate ground truth for training. Unlike these methods, RDVFI introduces a robust motion representation by predicting coefficients for high-order continuous pixel trajectories, thereby enabling a more expressive, physically consistent motion space.

### 2.3 DIFFUSION ACCELERATION

Accelerating diffusion inference is critical for practical VFI. Current techniques include rectified flow (Liu et al., 2022; 2023), adversarial training (Lin et al., 2025; Zhang et al., 2024), and score

Figure 3: Overall framework of the proposed RDVFI. Our RDVFI interpolates intermediate frames $\hat{I}_{vfi}$ in two phases: continuous pixel trajectories estimation and frame synthesis. **Phase 1** determines the coefficients of high-order continuous pixel trajectories, which index pixel offsets at arbitrary intermediate timestamps, using an LVDM. **Phase 2** samples the estimated trajectories for optical flows that warp input pixels to each interpolation timestamp. RDVFI then fuses the warped frames for interpolated frames with fine-grained details using the frame synthesis module.

distillation (Wang et al., 2023; Yin et al., 2024), to guide a lightweight, fewer-step student network to behave like a robust, multi-step teacher model. While one-step generation has been achieved in text-to-image (Liu et al., 2023; Yin et al., 2024) and video synthesis (Lin et al., 2025; Zhang et al., 2024), these distillation-based methods are difficult to adapt to VFI for two reasons. First, existing conditional frame-generation LVDMs may suffer from fewer diffusion sampling steps, aggravating their artifacts for large and complex motions. Second, the auxiliary networks required for distillation introduce significant training overhead. To this end, RDVFI does not involve distillation techniques. We demonstrate that by re-tasking the LVDM to predict motion trajectories rather than raw pixels, one-step sampling is inherently sufficient to resolve motion ambiguity while maintaining high-fidelity results through warping.

## 3 METHOD

We introduce RDVFI, a framework designed to bridge the gap between generative fidelity and motion coherence in VFI. We first outline the overall framework in Section 3.1. This is followed by our continuous pixel trajectory modeling in Section 3.2, the one-step LVDM in Section 3.3, the synthesis network in Section 3.4, and finally our two-stage training strategy in Section 3.5.

### 3.1 OVERALL FRAMEWORK

Given two endpoint frames $I_0, I_1$, RDVFI synthesizes a sequence of intermediate frames $\{\hat{I}_{\tau_j}\}_{j=1}^{L}$ at arbitrary timestamps $\tau_j \in (0, 1)$. The pipeline operates in two distinct phases: continuous pixel trajectory estimation at a fixed, relatively lower resolution, such as $448 \times 256$, and high-resolution frame synthesis (*e.g.*, at $1024 \times 576$), as shown in Fig. 3.

In the first phase, a Latent Video Diffusion Model (LVDM) generates a set of sparse latent keyframes $\{\hat{z}_{k_i}\}_{i=1}^{N}$ via one-step sampling. These keyframes serve as temporal anchors to determine the coefficients of continuous pixel trajectories. Because the LVDM operates at a reduced spatio-temporal resolution and models motion instead of texture, it achieves significant computational efficiency. In the second phase, RDVFI samples the estimated trajectories at each interpolation timestamp $\{\tau_j\}_{j=1}^{L}$, and we derive dense optical flows for each intermediate frame. These generated optical flows are used to warp the input frames, following a frame synthesis module to fuse and produce the final high-resolution output $\hat{I}_{vfi}$.

### 3.2 CONTINUOUS PIXEL TRAJECTORIES

We define continuous pixel trajectories as functions $\{f_{0 \to \tau}, f_{1 \to \tau}\}$ that map pixel offsets from $I_0$ or $I_1$ to any $\tau \in (0, 1)$. To ensure accuracy across large displacements, we decompose complex motions between input frames into small and easy-to-estimate components and solve them iteratively, as shown in Fig. 4. To this end, we first estimate motions across consecutive frames by:

$$f_{k_i \to k_{i+1}} = \phi_1(\hat{z}_{k_i}, \hat{z}_{k_{i+1}}). \tag{1}$$

Figure 4: The proposed continuous pixel trajectories estimator. The proposed RDVFI decomposes the large and complex motions between intermediate and input frames into small and simple motions across consecutive frames guided by latent keyframes generated by the LVDM. RDVFI then recursively fuses these motion components to compute optical flows between sparse latent keyframes and input frames. RDVFI fits a high-order continuous pixel trajectory functions using estimated optical flows, which can flexibly sample optical flows to warp input pixels to any intermediate time.

The estimated motion components are fused recursively to compute optical flows between each keyframe and input frames, following:

$$f_{0 \to k_{i+1}} = \phi_2(f_{0 \to k_i}, f_{k_i \to k_{i+1}}, z_0, \hat{z}_{k_{i+1}}). \tag{2}$$

In this formulation, $\phi_1$ and $\phi_2$ are convolutional neural networks in the supplementary Section B.2. We estimate the backward continuous pixel trajectories using the same network and weights, but in a reversed direction from $\tau = 1$ to $\tau = 0$. In addition to latent keyframes $\{k_i\}_{i=1}^N$, we also define $k_0, k_{N+1}$ to be encoded latent input frames and calculate the continuous pixel trajectories from $i = 0$ and $i = N + 1$ to $i = N$ and $i = 1$, respectively. This iterative refinement only updates a small and easy-to-estimate pixel offsets to previous flow results, allowing RDVFI to resolve large-scale motions more effectively and efficiently than direct-estimation baselines. Finally, we fit these keyframe optical flows using cubic convolution (Keys, 2003) to provide optical flows for all interpolation.

## 3.3 ONE-STEP LATENT VIDEO DIFFUSION MODEL

The continuous pixel trajectories introduced above are computed from keyframes, and we use the Latent Video Diffusion Model (LVDM) to generate those keyframes.

To illustrate that, we first revisit the preliminaries of LVDM. LVDMs generate video frames from noise based on input conditions in a low-resolution latent space, following a VAE decoder $\mathcal{D}(\cdot)$ to synthesize high-resolution video frames. In VFI, the LVDM first encodes the two input frames, $I_0$ and $I_1$, into the latent frames $z_0$ and $z_1$. These latent frames are then used as generation conditions. To learn the generation process, the LVDM degrades ground truth latent frames $z$ with noise $\epsilon^t \sim \mathcal{N}(0, I)$, generating noisy latent frames by $z^t = \alpha^t z + \sigma^t \epsilon^t$. Here, $\alpha^t$ and $\sigma^t$ are parameters determining how much noise is added. The diffusion timestep $t \in (0, T)$, with $t = T$, corresponds to pure noise. The LVDM uses a denoising network $\mathcal{F}_\theta(z^t; t, z_0, z_1)$ to estimate the added noise. The training objective is:

$$\mathcal{L}_\theta(t) = ||\mathcal{F}_\theta(z^t; t, z_0, z_1) - v^t||, \tag{3}$$

where $v^t = \alpha^t \epsilon^t - \sigma^t z$ is called the velocity of diffusion. After convergence, the LVDM recursively denoises randomly-sampled noise to generate noise-free intermediate latent frames $\hat{z}$. Because these LVDMs directly generate intermediate frames, they require tedious diffusion sampling to capture the fine-grained details of each frame, resulting in a high computation cost that is difficult to accelerate.

Unlike previous methods Zhu et al. (2024); Wang et al. (2024; 2025) that utilize the multi-step LVDM to generate all interpolated frames, our RDVFI only uses one-step LVDM to generate a few keyframes in a continuous pixel trajectory, resulting in significant acceleration. Our design is based on the fact that input frames in VFI have already provided the most desired intermediate appearances, and the core task is determining how to extract pixels from these input frames to generate intermediate frames. Because our LVDM only determines motions, it does not require dense diffusion sampling for fine-grained frame details, where one-step diffusion sampling can generate reasonable results. Furthermore, because the number of keyframes $N$ is often much smaller than the number of interpolated frames $L$, and generated motions can be flexibly resized to different resolutions, our LVDM can sample at a **fixed and relatively smaller** spatial-temporal frame resolution. This yields further acceleration and facilitates diffusion training. Please find the supplementary Section B.1 for the network details.

---

**Algorithm 1:** RDVFI Frame synthesis.

---

**Require:** Input frames $I_0$ and $I_1$, interpolation time $\tau = \{\tau_j\}_{j=1}^L$, estimated continuous pixel
      trajectories $\{f_{0\to\tau}, f_{1\to\tau}\}$
**Ensure:** Interpolated frames $\{\hat{I}_j\}_{j=1}^L$
1: **for** j=1, 2, ..., L **do**
2:    Sample optical flows $\{f_{0\to\tau_j}, f_{1\to\tau_j}\}$ from $\{f_{0\to\tau}, f_{1\to\tau}\}$
3:    Warp input frames $I_{0\to\tau_j} = Warp(I_0, f_{0\to\tau_j}), I_{1\to\tau_j} = Warp(I_1, f_{1\to\tau_j})$
4:    Find the nearest keyframe timestep, satisfying $k_i < \tau_j < k_{i+1}, k_0 = 0, k_{l+1} = 1$
5:    Sample optical flows $\{f_{\tau_i\to\tau_j}, f_{\tau_{i+1}\to\tau_j}\}$ from $\{f_{0\to\tau}, f_{1\to\tau}\}$
6:    Warp latent keyframes $z_{k_i\to\tau_j} = Warp(z_{k_i}, f_{k_i\to\tau_j}), z_{k_{i+1}\to\tau_j} = Warp(z_{k_{i+1}}, f_{k_{i+1}\to\tau_j})$
7:    $\hat{I}_j = \phi^s(I_0, I_1, I_{0\to\tau_j}, I_{1\to\tau_j}, f_{0\to\tau_j}, f_{1\to\tau_j}, z_{k_i\to\tau_j}, z_{k_{i+1}\to\tau_j})$
8: **end for**

---

### 3.4 FRAME SYNTHESIS

Given the estimated pixel trajectory introduced above, RDVFI finally generates all high-resolution
dense intermediate frames. As shown in Algorithm 1, RDVFI first samples the estimated continu-
ous pixel trajectories $\{f_{0\to\tau}, f_{1\to\tau}\}$ at different interpolation times $\tau = \{\tau_j\}_{j=1}^L$ to generate optical
flows $\{f_{0\to\tau_j} f_{1\to\tau_j}\}_{j=1}^L$. RDVFI then warps input frames $I_0, I_1$ with sampled optical flows, pro-
ducing bi-directional warped frames $\{I_{0\to\tau_j}, I_{1\to\tau_j}\}$. Unlike previous non-generative methods that
directly fuse the warped frames, we further incorporate latent keyframes generated through one-
step LVDM sampling to provide appearance changes that cannot be modeled by optical flows, such
as dynamic textures, lightning changes, and occlusions. For this target, we sample optical flows
$\{f_{k_i\to\tau_j}, f_{k_{i+1}\to\tau_j}\}$ between the latent keyframes $z_{k_i}, z_{k_{i+1}}$ that satisfy $k_i \leq \tau_j \leq k_{i+1}$. We further
warp these latent frames to time $\tau = \tau_j$, producing warped latent frames $\{z_{k_i\to\tau_j}, z_{k_{i+1}\to\tau_j}\}$. RD-
VFI then fuses warped frames $\{I_{0\to\tau_j}, I_{1\to\tau_j}\}$ and latent keyframes with a neural network $\phi^s(\cdot)$ to
produce the interpolated frames. Please find the supplementary Section B.3 for network details.

### 3.5 TRAINING PROCESS

As shown in Fig. 8, we train our RDVFI in two stages: motion-guided latent-to-frame generation
and one-step diffusion training.

**Motion-guided Decoding** RDVFI trains the continuous pixel trajectories estimator to predict ac-
curate optical flows for interpolation using given latent keyframes. Due to the lack of accurate
ground truth, RDVFI learns this process in an unsupervised manner with the loss on interpolated
frames. Existing research (Xue et al., 2019) has demonstrated that these task-oriented optical flows
can enhance downstream tasks and have been widely used in recent VFI methods (Jiang et al., 2018;
Bao et al., 2019; Xu et al., 2019; Liu et al., 2020; Huang et al., 2022; Reda et al., 2022; Li et al.,
2023; Danier et al., 2024). Specifically, RDVFI first selects $N$ keyframes $I^{\downarrow(t)}$ by frame skipping
and then downsamples selected keyframes to a relatively low resolution $I^{\downarrow(s,t)}$. RDVFI uses a
motion-guided decoding network to synthesize high-resolution frames $\hat{I}_{rec}$ from encoded ground
truth latent keyframes $z$. The training objective is formulated as follows:

$$\mathcal{L}_{rec}(\hat{I}_{rec}, I) = w_1 L_{lpips}(I, \hat{I}_{rec}) + w_2||I - \hat{I}_{rec}||_2, \tag{4}$$

In this formulation, $w_1$ and $w_2$ denote the loss weights, which are set to 0.2 and 1, respectively.

**One-step Diffusion Training** In this stage, RDVFI trains the LVDM to generate sparse latent
keyframes for interpolation. To achieve this, the continuous pixel trajectories estimator and synthesis
network are frozen, and only the parameters in the LVDM are updated. Rather than requiring the
LVDM to generate noise-free latent keyframes directly from pure noise, we train the LVDM with
randomly sampled $t \in (0, T)$ for higher stability. We linearly increase the diffusion timestep $t$
by adjusting the lower bound for random sampling according to the number of training steps. In
addition to the original pixel reconstruction loss in the latent space (Equation 3), the reconstruction
loss on the interpolated frames (Equation 4) is also employed, resulting in the following:

$$\mathcal{L}_{vfi}(I, \hat{I}_{vfi}) = \lambda_1 \mathcal{L}_\theta(t) + \lambda_2 \mathcal{L}_{rec}(\hat{I}_{vfi}, I). \tag{5}$$

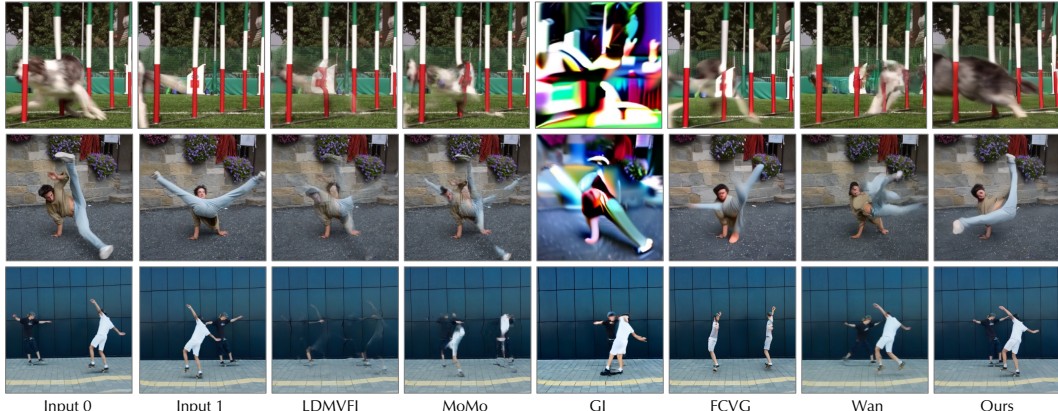

Figure 5: Visual comparison results on DAVIS-7 (Jain et al., 2024) (the top row), FCVG (Zhu et al., 2024) (the middle row), and our self-collected evaluation dataset (the bottom row). Our RDVFI-D method consistently outperforms existing diffusion-based VFI methods with less blurring, ghosting effects, color shifts, and fractions.

In this context, $\lambda_1$ and $\lambda_2$ are coefficients that balance each loss term, set to $1.0$ and $0.5$, respectively. Since our motion-guided decoding process involves continuous pixel trajectories estimation, the reconstruction loss term can serve as an implicit motion regularization term in the diffusion training and is essential for generating high-quality frames.

# 4 EXPERIMENTAL RESULTS

## 4.1 IMPLEMENTATION DETAILS

To verify that our RDVFI can adopt different pretrained LVDMs, we trained two versions: the Unet-based RDVFI-U, which builds upon 1.5B SVD (Blattmann et al., 2023), and the DiT-based RDVFI-D based on Wan2.1-Fun-1.3B-InP (Wang et al., 2025). Both of them use seven keyframes. During the training, we randomly resize images to one of four resolutions: $(448 \times 256, 576 \times 320, 1024 \times 576, 1280 \times 720)$. The LVDM and the continuous pixel trajectories estimator operate at the $448 \times 256$ resolution by resizing input images. The motion-guided decoding module then generates intermediate frames at the input resolutions with up-scaled optical flows. We use a fixed learning rate of 2e-5 for all experiments. We adopt eight A100-80G GPUs and a total batch size of 8. We train for 400K iterations in the first stage and 800K in the second. Please find the supplementary Section C for more details.

## 4.2 BENCHMARKING

**Selected Methods and Setting**  We compare the proposed RDVFI with several state-of-the-art methods, including conventional non-generative ones, such as RIFE (Huang et al., 2022), FILM (Reda et al., 2022), and AMT (Li et al., 2023), as well as the diffusion-based ones, including LDMVFI (Danier et al., 2024), MoMo (Lew et al., 2025), GI (Wang et al., 2024), FCVG (Zhu et al., 2024), and Wan 1.3B InP (Wang et al., 2025). Because methods may require different training strategies for optimal performance, we utilize the released weights without further tuning, as the authors have already best optimized them. We evaluate Wan (Wang et al., 2025) with blank text input. We do not compare with any distillation-based VFI method because current methods still focus on solving large and complex motions, and we have not found a baseline method that specifically involves VFI distillation.

**Results**  We report comparison results between the proposed RDVFI and baseline methods in Table 1, Fig. 6, and Fig. 5. The results show that our RDVFI outperforms existing baseline methods across the reported benchmark datasets. Our DiT-based method RDVFI-D outperforms the SVD-based version, RDVFI-U, benefiting from a more advanced backbone network. Image diffusion-based interpolation methods (Danier et al., 2024; Lew et al., 2025) cannot create complex motions between intermediate and input frames, thus producing severe ghosting effects. Directly generating

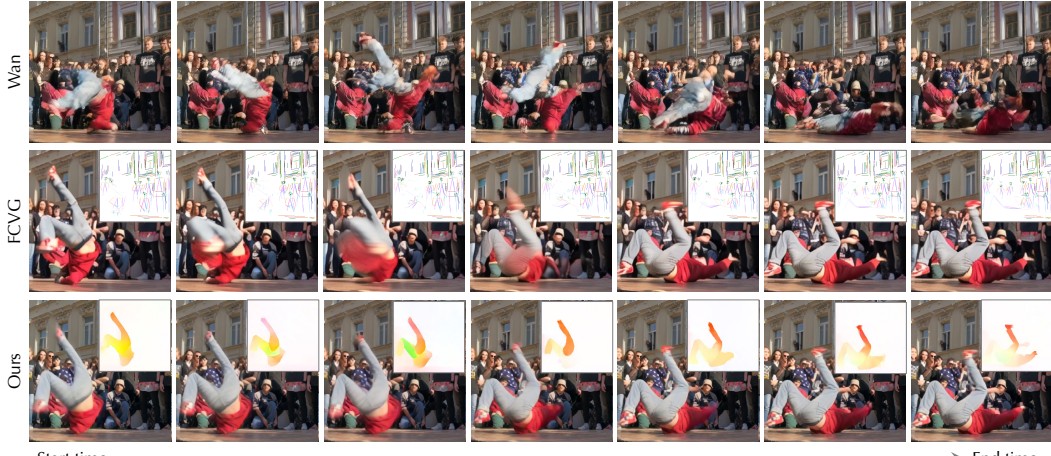

Figure 6: Sequence interpolation comparison. Our RDVFI-D can correctly estimate the continuous pixel trajectories between input frames from diffusion outputs and thus interpolate with correct motion and superior visual quality, compared with the direct generation method Wan (Wang et al., 2025) and the linearly-controlled FCVG (Zhu et al., 2024).

Table 1: Numeric comparison results on three benchmark datasets :DAVIS-7 (Jain et al., 2024), FCVG (Zhu et al., 2024) and our Pixels evaluation set. The best and the second-best results are highlighted by **bold** and underline. The proposed RDVFI outperforms existing baseline methods on most metrics.

| Methods | DAVIS-7 (Jain et al., 2024) | | | FCVG (Zhu et al., 2024) | | | Pixels | | |
|---|---|---|---|---|---|---|---|---|---|
| | LPIPS↓ | FID↓ | FVD↓ | LPIPS↓ | FID↓ | FVD↓ | LPIPS↓ | FID↓ | FVD↓ |
| Non-Generative Methods | | | | | | | | | |
| AMT (Li et al., 2023) | 0.254 | 34.65 | 234.50 | 0.224 | 44.74 | 375.00 | 0.361 | 41.36 | 378.86 |
| RIFE (Huang et al., 2022) | 0.258 | 23.98 | 240.04 | 0.247 | 39.01 | 366.14 | 0.278 | 31.10 | 207.82 |
| FILM (Reda et al., 2022) | 0.271 | 30.16 | 214.80 | 0.241 | 39.82 | 279.08 | 0.251 | 27.06 | 158.68 |
| GIMM-VFI (Guo et al., 2024) | 0.276 | 24.10 | 245.02 | 0.231 | 38.71 | 292.08 | 0.269 | 29.21 | 192.37 |
| Image Diffusion-based Methods | | | | | | | | | |
| LDMVFI (Danier et al., 2024) | 0.276 | 22.10 | 245.02 | 0.228 | 37.74 | 371.49 | 0.287 | 31.36 | 211.70 |
| MoMo (Lew et al., 2025) | 0.268 | 23.67 | 240.09 | 0.207 | 33.59 | 261.37 | 0.269 | 27.31 | 230.19 |
| Video Diffusion-based Methods | | | | | | | | | |
| TRF (Feng et al., 2024) | 0.270 | 29.12 | 230.12 | 0.331 | 45.37 | 305.88 | 0.301 | 35.31 | 279.65 |
| GI (Wang et al., 2024) | 0.267 | 27.71 | 211.47 | 0.334 | 43.08 | 282.22 | 0.273 | 33.27 | 251.38 |
| FCVG (Zhu et al., 2024) | 0.266 | 25.96 | 207.17 | 0.266 | 31.24 | 225.48 | 0.257 | 24.51 | 137.57 |
| Wan (Wang et al., 2025) | 0.323 | 26.97 | 248.14 | 0.223 | 28.52 | 214.45 | 0.261 | 22.48 | 131.22 |
| Our RDVFI-U | 0.260 | 23.65 | 201.49 | 0.220 | 27.03 | 217.72 | 0.261 | 23.71 | 129.33 |
| Our RDVFI-D | **0.251** | **21.17** | **189.37** | **0.201** | **19.98** | **197.86** | **0.253** | **19.48** | **119.21** |

intermediate frames with Latent Video Diffusion Models (LVDM) may introduce severe degradations (Wang et al., 2024), due to the mismatched interpolation results from separate forward and backward interpolation from two image-to-video SVD (Blattmann et al., 2023) models; FCVG (Zhu et al., 2024) attempts to mitigate such misalignment by involving linear motion controls, however, sacrificing motion generation ability of VDMs and utilizing them as shaders. As shown in Fig. 6, FCVG (Zhu et al., 2024) produces severe degradations when the matching-based linear motion controller does not distinguish the falling boy. Sharing the same backbone, however, our RDVFI-D significantly outperforms Wan (Wang et al., 2025). Our motion-guided decoding pipeline generates intermediate frames by warping, which forces the diffusion model to create latents that can accurately restore motions across frames. Thus, our interpolation is smoother and more stable, producing the fewest artifacts compared to other LVDM-based solutions. Please find our supplementary video demo and Section D for more dynamic and static visual comparisons.

## 4.3 EFFICIENCY COMPARISONS

We report detailed efficiency metrics in Table 2. As we can observe, our RDVFI is the fastest diffusion-based interpolation method, which can interpolate at real time (17 FPS) at the resolution of $1024 \times 576$. Although a non-generative baseline method, RIFE (Huang et al., 2022), is slightly

Table 2: We compare the inference efficiency between our RDVFI and existing interpolation baseline methods, including AMT (Li et al., 2023), RIFE (Huang et al., 2022), FILM (Reda et al., 2022), LDMVFI (Danier et al., 2024), MoMo (Lew et al., 2025), GI (Wang et al., 2024), FCVG (Zhu et al., 2024), and Wan (Wang et al., 2025), on the FCVG (Zhu et al., 2024) dataset for $\times 24$ interpolation at the $1024 \times 576$ resolution. We **bold** and underline the best and the second-best results, respectively. Our RDVFI is the fastest and most memory-efficient diffusion-based interpolation method. Although RIFE (Huang et al., 2022) is slightly faster than our RDVFI, our RDVFI outperforms it with a clear margin for challenging motions.

| Method | GPU Mem. (GB) | Runtime (sec.) | FID↓ | FVD↓ |
|---|---|---|---|---|
| Non-Generative Methods | | | | |
| AMT Li et al. (2023) | 13.5 | 0.210 | 44.74 | 375.00 |
| RIFE Huang et al. (2022) | **1.4** | **0.025** | 39.01 | 366.14 |
| FILM Reda et al. (2022) | 8.0 | 0.830 | 39.82 | 279.08 |
| Image Diffusion-based Methods | | | | |
| LDMVFI Danier et al. (2024) | 21.7 | 1.563 | 37.74 | 371.49 |
| MoMo Lew et al. (2025) | 3.9 | 0.157 | 33.59 | 261.37 |
| Video Diffusion-based Methods | | | | |
| GI Wang et al. (2024) | 23.5 | 29.613 | 45.37 | 282.22 |
| FCVG Zhu et al. (2024) | 27.6 | 14.381 | 31.24 | 225.48 |
| Wan Wang et al. (2025) | 18.0 | 2.579 | 28.52 | 214.45 |
| Our RDVFI-U | 14.2 | 0.137 | 27.03 | 217.72 |
| Our RDVFI-D | 13.1 | 0.057 | **19.98** | **197.86** |

Table 3: Ablation study on different resolutions (3a) and network designs (3b).

(a) Because we perform the diffusion sampling at fixed and relatively smaller resolution than the interpolated frame, the runtime of the algorithm does not change a lot with the resolution increase.

(b) Our RDVFI-D consistently outperforms all ablation experiments, showing the best interpolation accuracy in terms of both reconstruction and perceptual metrics.

| Resolution | Runtime (ms) | |
|---|---|---|
| | One-step Wan | RDVFI-D |
| $576 \times 320$ | 103.3 | 55.3 |
| $1024 \times 576$ | 267.7 | 58.01 |
| $1280 \times 720$ | 314.3 | 63.49 |

| Setting | LPIPS↓ | FID↓ | FVD↓ |
|---|---|---|---|
| Direct Warping | 0.287 | 42.33 | 327.11 |
| L loss | 0.269 | 33.29 | 281.37 |
| L+P-L2 loss | 0.251 | 29.37 | 267.42 |
| RDVFI-D | 0.224 | 23.71 | 209.38 |

faster than our RDVFI, it cannot deal with large complex motions and introduces severe visual degradations. In contrast, our RDVFI can efficiently and effectively interpolate large complex motions, outperforming RIFE with a clear margin, as shown in Table 1, Fig. 5, and Fig. 6.

## 4.4 ABLATION STUDY

**Motion-guided Decoding Pipeline** We compare the runtime of one-step Wan (Wang et al., 2025), which directly reduces the sampling step of the pretrained Wan model, and our RDVFI-D. These two models share the same diffusion backbone and sampling step with two core differences. First, Wan's resolution changes with interpolated frames, whereas our RDVFI-D operates at a fixed, relatively lower resolution. Second, Wan generates high-resolution frames with the VAE decoder and our RDVFI-D scales up the estimated optical flows and warps high-resolution inputs for interpolation. In addition, we claim that iterative motion fusion for continuous pixel trajectories estimation is one of our key contributions to accurate motion estimation. Thus, we degrade our motion-guided decoding pipeline by removing iterative flow fusion and estimating pixel offsets between each keyframe and the input frames from scratch via regression. We define this setting as "Direct Warping".

**Training Strategy** To validate our latent-pixel training strategy, we degrade our training objective by removing the pixel-space perceptual loss only, formatting"L+P-L2 loss", and all pixel-wise loss, defined as "L loss".

As shown in Table 3a and Table 3b, our RDVFI-D consistently outperforms all degraded versions on efficiency or accuracy. Benefiting from our motion-guided decoding, RDVFI-D can interpolate at different resolutions at the same diffusion sampling cost, resulting in a significant efficiency gain compared with traditional diffusion networks, as in Table 3a. Table 3b also shows the effectiveness of our flow fusion strategy and training objective design. For fair comparison, we fine-tune

Table 4: Human evaluation results. Most viewers select our RDVFI-D as the best-performed interpolation method when compared with existing diffusion-based state-of-the-art methods in terms of both generated motion, frame details, and overall quality. Our RDVFI-D surpasses the second-best method, Wan (Wang et al., 2025), for 60% more.

| Direction | LDMVFI | MoMo | GI | FCVG | Wan | Ours |
|-----------|--------|------|-----|------|-----|------|
| Motion | 0.9% | 0.8% | 1.1% | 4.1% | 16.2% | **76.9%** |
| Details | 0.3% | 0.9% | 1.5% | 9.2% | 13.7% | **74.4%** |
| Overall | 0.5% | 0.6% | 1.0% | 8.0% | 14.2% | **75.7%** |

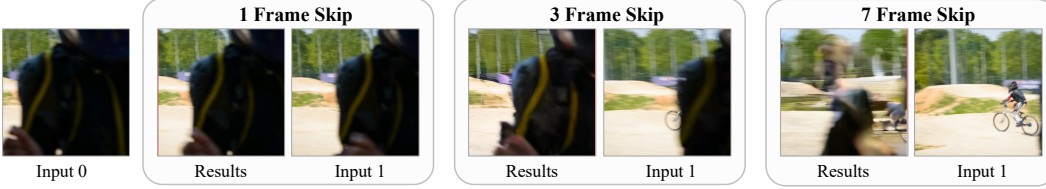

Figure 7: Our RDVFI can robustly interpolate when input frames have consistent objects and enough overlapping regions, such as the 1-frame and 3-frame skip cases. However, when the overlapped regions become minimal, as the ending input frame is produced by 7 frames skipped, the algorithm cannot generate coherent and accurate intermediate optical flows to warp input pixels for interpolation, resulting in severe degradations.

each ablation experiment and report our results at the same iterations. Please find Section A for the ablation study on the keyframe number.

## 4.5 HUMAN EVALUATION

To better illustrate the visual gains of the proposed RDVFI over existing solutions, we also include a human evaluation study involving 25 participants. We form a questionnaire with ten groups of videos, each group contains six video clips from the proposed RDVFI-D and five state-of-the-art methods, including Wan (Wang et al., 2025), FCVG (Zhu et al., 2024), GI (Wang et al., 2024), MoMo (Lew et al., 2025), and LDMVFI (Danier et al., 2024). The respondents are expected to select the best video clip from each group based on the generated motion quality, frame details, and overall quality. As shown in the Table 4, the proposed RDVFI-D significantly outperforms all state-of-the-art methods, with more than 75% of respondents selecting our RDVFI-D as the best-performing method, surpassing the second-highest method, Wan (Wang et al., 2025), by around 5 times.

## 4.6 FAILURE CASE AND LIMITATIONS

Our method warps input frames with generated intermediate optical flows for interpolation, thus requiring the input frames to contain consistent objects and enough overlapping regions. As shown in Fig. 7, with the same starting frame, our method can robustly interpolate with the ending input frame produced by 1 frame and 3 frames skipped. However, when the overlapped regions become minimal, even with a changed main object in the 7-frame-skipped situation, the algorithm struggles to generate coherent and accurate optical flows to move pixels from the inputs to each intermediate frame, resulting in severe degradations through interpolation.

## 5 CONCLUSIONS

In this paper, we propose the first real-time video diffusion-based Video Frame Interpolation (VFI) pipeline that can run 17FPS at $1024 \times 576$ resolution with even superior interpolation quality than current multi-step solutions. Our work is advancing diffusion-based VFI to more practical and challenging scenarios, such as super slow motion and video compression. This work also hopes to suggest that solving ambiguous components with diffusion models, rather than end-to-end generation, may lead to superior accuracy and efficiency.

ACKNOWLEDGEMENTS

This work is partially supported by the RGC Early Career Scheme (ECS) No. 24209224. This work was done at Yongrui Ma's internship at the Bytedance Inc.

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
