# OpenReview forum: "Realtime Video Frame Interpolation using One-Step Diffusion Sampling"
_ICLR.cc/2026/Conference — ICLR 2026 Poster_

### Official Review · Reviewer_1Tgm · 2025-10-27

**Soundness:** 3
**Presentation:** 3
**Contribution:** 3
**Rating:** 4
**Confidence:** 5

**Summary:**

This work introduces an efficient video interpolation approach based on a video diffusion model, enabling real-time frame interpolation. The proposed pipeline consists of three main stages: (1) estimating sparse, low-resolution keyframes through a one-step video diffusion process; (2) extracting a complex motion field from these keyframes using a continuous motion estimator; and (3) synthesizing full-resolution intermediate frames from the motion field and the input frames ($I_0$ and $I_1$) using a frame synthesis network. In addition, the authors propose a two-stage training framework to address the unstable convergence may observed in simple end-to-end training.

**Strengths:**

Traditional non-generative methods preserve the appearance of input frames well but struggle to interpolate dynamic motions. In contrast, diffusion-based methods effectively model and interpolate complex motions but are computationally expensive and often fail to maintain the appearance consistency of input frames.

This paper proposes a novel approach that disentangles motion from the video diffusion model (VDM) and applies the estimated sparse motion to achieve efficient and effective video interpolation.
The main strengths of the paper can be summarized as follows:

1) Disentangling motion field from a one-step VDM is both novel and effective.

2) Integrating the motion field extracted from the VDM into a frame synthesis framework improves efficiency compared to full VDM-based methods.

3) The qualitative and quantitative results are impressive and demonstrate the effectiveness of the proposed method.

**Weaknesses:**

1) Insufficient experimental analysis on efficiency:
Although the proposed framework emphasizes efficiency as a key contribution, the paper lacks detailed quantitative evidence. The authors should include explicit efficiency metrics in Table 1, such as inference time (s/frame) and memory usage (VRAM in GB), to substantiate their claims.

2) Lack of detailed baseline categorization:
For video interpolation methods based on video diffusion models (VDMs), it is crucial to clearly categorize baselines into zero-shot, fine-tuned, and fully trained settings. Without this distinction, the fairness and clarity of the comparison become ambiguous, potentially undermining the paper’s original motivation. Reporting the total training compute—e.g., approximate FLOPs or other comparable measures—would further improve the transparency and fairness of the evaluation.

3) Limited ablation studies:
The initial estimation of $k$ keyframes using the VDM plays a central role in determining the motion field and thus in the overall effectiveness of the proposed method. However, the paper lacks sufficient analysis of how the number of keyframes ($k$) affects both efficiency and interpolation quality. A detailed ablation study on this factor would greatly enhance the understanding of the method’s underlying principles.

**Questions:**

1) Please include explicit efficiency metrics — such as inference time (s/frame), memory usage (GB), and training compute (e.g., approximate FLOPs or other comparable measures) — in the main manuscript to better support the claimed efficiency of the proposed framework.

2) Please clearly categorize the baselines into zero-shot, fine-tuned, and fully trained settings for a fair comparison. Additionally, consider including more zero-shot video diffusion interpolation works such as TRF [1] and ViBiDSampler [2], which are missing from the current submission.

3) If possible, add a brief ablation study on the number of keyframes ($k$). This would significantly improve the clarity and completeness of the paper by illustrating how $k$ influences both interpolation quality and computational efficiency.

4) It would be helpful to include failure case analysis. Presenting representative failure cases and describing in which situations such samples frequently occur would provide deeper insight into the limitations and behavior of the proposed method.

[1] Explorative Inbetweening of Time and Space, ECCV 2024.

[2] ViBiDSampler: Enhancing Video Interpolation Using Bidirectional Diffusion Sampler, ICLR 2025.

---

> ### Author Response · Authors · 2025-11-25
>
> ## More Detailed Experiments
> We report efficiency comparisons and categories of video diffusion-based methods as follows. Please find Table 2 in the revised paper.
>
> Although the non-generative interpolation method, RIFE, can interpolate slightly faster than our RDVFI, it cannot handle large, complex motions, resulting in severe degradations and artifacts. In contrast, as the fastest diffusion-based interpolation method, our RDVFI can effectively interpolate large, complex motions, consistently outperforming existing interpolation methods by a clear margin.
>
>
>
> ### Non-generative based interpolation methods
> | Method | LPIPS | FID | FVD | Memory (GB) | Runtime (sec./frame) |
> | --- | --- | --- | --- |-------------|----------------------|
> | AMT[1] | 0.224 | 44.74 | 375.00 | 13.5        | 0.210                |
> | RIFE[2] | 0.247 | 39.01 | 366.14 | ***1.4***   | ***0.025***          |
> | FILM[3] | 0.241 | 39.82 | 279.08 | 8.0         | 0.830                |
>
> ### Image diffusion based interpolation methods
> | Method | LPIPS | FID | FVD | Memory (GB) | Runtime (sec./frame) |
> | --- | --- | --- | --- |-----------|----------------------|
> | LDMVFI[4]| 0.228 | 37.74 | 371.49 | 21.7      | 1.563                |
> | MoMo[5] | 0.207 | 33.59 | 261.37 | ***3.9***       | ***0.157***                |
>
> ### Video diffusion based interpolation methods
> | Method | Category            | LPIPS | FID   | FVD    | Memory (GB) | Runtime (sec./frame) |
> | --- |---------------------|-------|-------|--------|-------------|----------------------|
> | TRF[6] | zero-shot | 0.331 | 45.37 | 305.88 | 13.3        | 7.382                |
> | ViBiDSampler[7] | zero-shot | 0.292 | 39.83 | 257.15 | 26.24 | 3.708 |
> | GI[8] | Fine-tuned          | 0.334 | 43.08 | 282.22 | 23.5        | 29.613               |
> | FCVG[9] | Fine-tuned          | 0.266 | 31.24 | 225.48 | 27.6        | 14.381               |
> | Wan[10] | Fully Trained | 0.223 | 28.52 | 214.45 | 18.0        | 2.579                |
> | Our RDVFI-U | Fine-tuned | 0.220       | 27.03       | 217.72       | 14.2        | 0.137                |
> | Our RDVFI-D | Fine-tuned | ***0.201*** | ***19.98*** | ***197.86*** | ***13.1***        | ***0.058***                |
>
> ## Ablation study on keyframes number
> We provide an ablation study on keyframe numbers in the FCVG dataset. Because RDVFI-D only interpolates 4N keyframes, we degrade the RDVFI-U due to its flexibility. As we can observe, as the keyframe number increases, the interpolation quality improves accordingly due to its growing ability to model intermediate motions across frames and more accurate optical flow estimation. The gain becomes moderate as the number increases, as more and more motions can be well-modeled and solved with the generated keyframes. Generating more frames with the diffusion model only introduces a marginal performance gain, which supports our claim that generating a few keyframes is sufficient for interpolation and results in superior efficiency.
>
> | Keyframes Number    | LPIPS | FID   | FVD    |
> |---------------------|-------|-------|--------|
> | 1                   | 0.231 | 33.46 | 297.71 |
> | 2                   | 0.227 | 31.19 | 272.31 |
> | 3                   | 0.224 | 29.83 | 257.49 |
> | 4                   | 0.225 | 28.37 | 242.08 |
> | 5                   | 0.224 | 27.22 | 229.91 |
> | 6                   | 0.221 | 27.10 | 220.24 |
> | 7 (Default setting) | 0.220 | 27.03 | 217.72 |
> | 8                   | 0.221 | 27.01 | 218.11 |
>
> ## Failure case analysis
> Our method warps input frames with generated intermediate optical flows for interpolation, thus requiring the input frames to contain consistent objects and enough overlapping regions. Thus, when the overlapped regions become minimal, our method would struggle to generate coherent and accurate optical flows to move pixels from the inputs to each intermediate frame, resulting in severe degradations through interpolation. Please find Section 4.6 in the revised paper for more details.

---

> > ### Author Response · Authors · 2025-11-25
> >
> > [1] Li, Zhen, et al. "Amt: All-pairs multi-field transforms for efficient frame interpolation." Proceedings of the IEEE/CVF Conference on Computer Vision and Pattern Recognition. 2023.
> >
> > [2] Huang, Zhewei, et al. "Real-time intermediate flow estimation for video frame interpolation." European Conference on Computer Vision. Cham: Springer Nature Switzerland, 2022.
> >
> > [3] Reda, Fitsum, et al. "Film: Frame interpolation for large motion." European Conference on Computer Vision. Cham: Springer Nature Switzerland, 2022.
> >
> > [4] Danier, Duolikun, Fan Zhang, and David Bull. "Ldmvfi: Video frame interpolation with latent diffusion models." Proceedings of the AAAI Conference on Artificial Intelligence. Vol. 38. No. 2. 2024.
> >
> > [5] Lew, Jaihyun, et al. "Disentangled motion modeling for video frame interpolation." Proceedings of the AAAI Conference on Artificial Intelligence. Vol. 39. No. 5. 2025.
> >
> > [6] Feng, Haiwen, et al. "Explorative inbetweening of time and space." European Conference on Computer Vision. Cham: Springer Nature Switzerland, 2024.
> >
> > [7] Yang, Serin, Taesung Kwon, and Jong Chul Ye. "Vibidsampler: Enhancing video interpolation using bidirectional diffusion sampler." arXiv preprint arXiv:2410.05651 (2024).
> >
> > [8] Wang, Xiaojuan, et al. "Generative inbetweening: Adapting image-to-video models for keyframe interpolation." arXiv preprint arXiv:2408.15239 (2024).
> >
> > [9] Zhu, Tianyi, et al. "Generative inbetweening through frame-wise conditions-driven video generation." Proceedings of the Computer Vision and Pattern Recognition Conference. 2025.
> >
> > [10] Wan, Team, et al. "Wan: Open and advanced large-scale video generative models." arXiv preprint arXiv:2503.20314 (2025).

---

### Official Review · Reviewer_G3iW · 2025-10-29

**Soundness:** 2
**Presentation:** 2
**Contribution:** 2
**Rating:** 4
**Confidence:** 4

**Summary:**

The authors propose a novel one-step diffusion method for video frame interpolation. The authors present a two-stage strategy. Specifically, it first decomposes the interpolation process into motion modeling and frame synthesis. The motion modeling and frame synthesis module are trained together in an end-to-end format, with the ground truth frame latents as the additional input. In the second training stage, they train the diffusion model for denoising the pseudo ground truth frame latents for motion prediction. In their experiments, the proposed method outperforms the baselines with fast speed.

**Strengths:**

- The authors propose a novel framework for video frame interpolation, achieving fast speed and competitive performance.
- The proposed continuous motion field representation enables more flexible motion modeling and generates plausible flow samples, as evidenced by the visualizations.

**Weaknesses:**

- The paper lacks visualizations on more challenging scenarios, such as the “breaking dance” case in the DAVIS dataset.
- The flow sampling mechanism is not sufficiently explained; more details on how samples are generated and utilized would strengthen the paper.
- The paper is missing discussions of closely related works that also integrate motion or optical flow modeling in video generation/interpolation, such as *VideoJAM* [1], *Motion-I2V* [2], and *GIMM-VFI* [3].
- Continuous temporal results. More visualizations like Figure 5 to present the continuous motion modeling and interpolation ability.

[1] VideoJAM: Joint Appearance-Motion Representations for Enhanced Motion Generation in Video Models.

[2] Motion-I2V: Consistent and Controllable Image-to-Video Generation with Explicit Motion Modeling.

[3] Generalizable Implicit Motion Modeling for Video Frame Interpolation.

**Questions:**

- Is it necessary to use the VAE encoder? Given the recent progress in the academy, it would be interesting to replace the VAE encoder with other encoders, such as DINOv2.
- Please add more visualizations for the cases indicated in the weakness section, including more challenging scenarios and continuous temporal results, to support the claimed interpolation ability.
- It is necessary to have discussions with previous closely related work to enhance the clarity of the paper's motivation and contribution.
- Please add more detailed descriptions for the core motion modeling part, especially the flow sampling operation, for better presentation.

---

> ### Author Response · Authors · 2025-11-25
>
> ## Discussion with Related Methods
> VideoJAM[1] and Motion-I2V[2] utilizes optical flows as motion priors to improve frame generation diffusion model. In constrast, our RDVFI solves the ambiguity of intermediate motions with a diffusion model, producing optical flows to warp input frames to generate intermediate frames.
>
> | Method                     | What Diffusion Generates                                  | How Optical Flows Work                                                               |
> |----------------------------|-----------------------------------------------------------|--------------------------------------------------------------------------------------|
> | VideoJAM[1], Motion-I2V[2] | Interpolated frames with fine-grained details.            | Work as a side information. Do not utilize the physical properties of optical flows. |
> | Our RDVFI | Ambiguous intermediate motions across consecutive frames. | Warp pixels from two input frames to generate each intermediate frame                |
>
> The GIMM-VFI[3] still follows the non-generative interpolation routine, scaling optical flows across input frames with a multilayer perceptron network. However, it is still regression-based and limited to solve challenging nonlinear and nonrigid motions. Differently, our RDVFI generates intermediate optical flows by diffusion sampling, resulting in superior interpolation quality.
>
> ## Flow sampling mechanism
> 1. As introduced in Section 3.2, our continuous motion field estimator first estimates the optical flows $\{f_{i\rightarrow i+1}, f_{i+1\rightarrow i}\}$ across consecutive keyframes and then iteratively fuses them to estimate the optical flows $\{f_{i\rightarrow 0}, f_{i\rightarrow N}\}$ between each keyframe and the input frames $\{I_0, I_N\}$.
> 2. We select a function $F(x,y,t)=(u,v)$ to model pixel movements across time, where $x,y$ are pixel locations at the input frames, $t$ is the time for interpolation, and $(u,v)$ are pixel offsets. The function can be polynomial, Bezier splines, or Natural B-Splines. We select the cubic convolution method[4] because all of these functions have similar performance, but the cubic convolution method can sample motions more efficiently.
> 3. We then use the estimated optical flows $\{f_{i\rightarrow 0}, f_{i\rightarrow N}\}$ to fit the function $F(x,y,t)$, and finally interpolate all intermediate frames with optical flows sampled from the function $F(x,y,t)$.
>
>
> ## Weakness discussion
> We provide additional discussions about the failure cases in Section 4.6 of the revised paper. Additionally, we also provide two visual examples in Section D of the appendix, which only contain small and simple motions. Non-generative methods can solve these motions as they can be well approximated by predetermined functions. Although both non-generative methods and our RDVFI can generate visually acceptable results, they can obtain even superior numeric metrics than our RDVFI, as they only focus on small and simple motions, while we deal with both small simple motions and large complex ones.
>
> ## More visualization
> Please find Figure 8 and Figure 9 in the revised paper.
>
> ## VAE Encoder
> We agree that recent encoders, such as DINO, can further improve interpolation accuracy as they are more robust and involve high-level semantic information. We will try them in our future work.
>
>
> [1] Chefer, Hila, et al. "Videojam: Joint appearance-motion representations for enhanced motion generation in video models." arXiv preprint arXiv:2502.02492 (2025).
>
> [2] Shi, Xiaoyu, et al. "Motion-i2v: Consistent and controllable image-to-video generation with explicit motion modeling." ACM SIGGRAPH 2024 Conference Papers. 2024.
>
> [3] Guo, Zujin, Wei Li, and Chen Change Loy. "Generalizable implicit motion modeling for video frame interpolation." Advances in Neural Information Processing Systems 37 (2024): 63747-63770.
>
> [4] Keys, Robert. "Cubic convolution interpolation for digital image processing." IEEE transactions on acoustics, speech, and signal processing 29.6 (2003): 1153-1160.

---

### Official Review · Reviewer_R4Dp · 2025-10-29

**Soundness:** 3
**Presentation:** 3
**Contribution:** 3
**Rating:** 6
**Confidence:** 4

**Summary:**

1. The core idea of RDVFI is to disentangle VFI into two stages: motion prediction and appearance generation. Generating new frames based on low-resolution motion information enables RDVFI to perform effectively in both inference speed and generation authenticity.
2. This is the first diffusion-based VFI method with one-step inference, which achieves 50× acceleration compared to SOTA with also better results.

**Strengths:**

1. In the diffusion stage, this method generates low-resolution optical flow as an intermediate result to both accelerate the diffusion process and improve the stability of interpolation.
2. Bi-directional interpolation significantly improves generation authenticity, making RDVFI get SOTA performance on DAVIS and FCVG benchmarks.
3. The video demo shows significantly better results than existing methods.

**Weaknesses:**

1. Optical flow is the key information for the entire pipeline, as it is utilized for both image warping and feature warping in a bidirectional manner. However, in the first training stage, optical flow results are trained in an unsupervised way. The rationality of this setting requires further verification.
2. The number of testsets is limited and lacks diversity in their sources. The authors evaluate their method on DAVIS-7 and FCVG, however FCVG is sampled from DAVIS and RealEstate10K.
3. For the RDVFI-U and RDVFI-D models, the authors set different numbers of key frames, but did not provide a reasonable explanation and lacked corresponding ablation experiments.

**Questions:**

Please refer to the Weaknesses section.

---

> ### Author Response · Authors · 2025-11-25
>
> ## Unsupervised Optical Flow
> As discussed in [1], task-oriented optical flows can benefit downstream tasks more than ground-truth optical flows. Instead of supervising estimated optical flows with ground truth, task-oriented optical flows that are calculated in an unsupervised manner can better fit the downstream task and lead to superior performance. As a common technique, recent flow-based video frame interpolation methods[2,3,4] train their flow estimation modules in this unsupervised manner, generating stable interpolation results, as our RDVFI method does.
>
> ## Limited Test Sets
> To evaluate the proposed RDVFI more completely, we further collect an evaluation benchmark from Pixels, consisting of 100 video clips. Please find Table 1, Figure 8, and Figure 9 for numeric and visual comparisons, respectively.
>
> ## Mismatched Keyframe Number
> We set 7 key frames in RDVFI-U as it balances computation and performance well, as shown in the following table for ablation study on keyframes in RDVFI-U, and match the x8 interpolation setting in the DAVIS-7 dataset. Differently, we set 8 key frames in RDVFI-D because Wan only interpolates 4N frame interpolation.
>
> | Keyframes Number | LPIPS | FID   | FVD    |
> |------------------|-------|-------|--------|
> | 1                | 0.231 | 33.46 | 297.71 |
> | 2                | 0.227 | 31.19 | 272.31 |
> | 3                | 0.224 | 29.83 | 257.49 |
> | 4                | 0.225 | 28.37 | 242.08 |
> | 5                | 0.224 | 27.22 | 229.91 |
> | 6                | 0.221 | 27.10 | 220.24 |
> | 7 (Default setting)               | 0.220 | 27.03 | 217.72 |
> | 8                | 0.221 | 27.01 | 218.11 |
>
> [1] Xue, Tianfan, et al. "Video enhancement with task-oriented flow." International Journal of Computer Vision 127.8 (2019): 1106-1125.
>
> [2] Huang, Zhewei, et al. "Real-time intermediate flow estimation for video frame interpolation." European Conference on Computer Vision. Cham: Springer Nature Switzerland, 2022.
>
> [3] Reda, Fitsum, et al. "Film: Frame interpolation for large motion." European Conference on Computer Vision. Cham: Springer Nature Switzerland, 2022.
>
> [4] Li, Zhen, et al. "Amt: All-pairs multi-field transforms for efficient frame interpolation." Proceedings of the IEEE/CVF Conference on Computer Vision and Pattern Recognition. 2023.

---

### Official Review · Reviewer_Ba4o · 2025-10-29

**Soundness:** 2
**Presentation:** 3
**Contribution:** 2
**Rating:** 2
**Confidence:** 5

**Summary:**

This paper proposes a video frame interpolation method using a one-step diffusion model, aiming to improve inference efficiency by eliminating the need for multi-step denoising. The approach decomposes intermediate frame generation into two stages: the first estimates a continuous motion field between the input frames, and the second synthesizes intermediate frames by warping the inputs according to the predicted motion field. Experimental results are compared against conventional video frame interpolation methods as well as diffusion-based approaches.

**Strengths:**

The paper aims to improve the efficiency of diffusion models for video frame interpolation by disentangling in-between frame synthesis into two stages: continuous motion prediction and warping. The continuous motion (which use a spline interpolation curve) prediction stage is lightweight, operating on latent features from spatially and temporally downsampled video representations, which reduces memory consumption. In addition, the method employs a one-step diffusion model to predict the latent features during inference time further increasing inference speed.

**Weaknesses:**

The main weakness is the experiment does not convince me of the effectiveness of the method:
1. There are only six qualitative video comparisons in the supplementary material, which are insufficient to demonstrate the superior quality of the proposed method. Moreover, I did not observe a clear visual difference between Wan and the proposed approach.
2.  Lacks baselines on direct one/few-step distillation of video in-betweening diffusion models, rather than decomposing the process into two stages as done in the paper.

**Questions:**

1. How does  the number of keyframes in the continuous motion representation affect the inbetweening results especially in 24x interpolation?
2. Directly fine-tuning the original diffusion denoiser to perform full noise removal in a single step seems rather ambitious. Incorporating a distillation-based loss might help improve stability and performance.

---

> ### Author Response · Authors · 2025-11-25
>
> ## Effectiveness
> Our ***RDVFI consistently outperforms existing video frame interpolation methods on numeric comparisons***, as shown in Table 1 in the manuscript. In addition, we are pleased to see that ***both reviewers*** R4Dp, G3iW, and 1Tgm agree that ***the proposed RDVFI achieves significant visual improvements over existing methods*** in video frame interpolation for large complex motions. More specifically, reviewer R4Dp claims that our video demo shows significantly better results than existing methods; reviewer G3iW states that the proposed RDVFI achieves fast speed and competitive performance; and reviewer 1Tgm asserts that our qualitative and quantitative results are impressive, demonstrating the effectiveness of the proposed RDVFI against Wan.
>
> To better illustrate the visual gain of the proposed RDVFI over existing solutions, we also include a human evaluation study. We form a questionnaire with ten groups of videos, each group contains six video clips from the proposed RDVFI and five state-of-the-art methods, including Wan, FCVG, GI, MoMo, and LDMVFI. The respondents are expected to select the best video clip from each group based on the generated motion quality, frame details, and overall quality. As shown in the following table, the proposed RDVFI significantly outperforms all state-of-the-art methods, with more than 75% of respondents selecting our RDVFI as the best-performing method, ***surpassing the second-highest method, Wan, by 60%.***
>
> | Method     | Motion Quality (\%) | Overall Quality (\%) |
> |------------|---------------------|----------------------|
> | ***Ours*** | ***76.9***          | ***75.7***           |
> | Wan        | 16.2                | 14.1                 |
> | FCVG | 4.1 | 8.0 |
> | GI | 1.1 | 1.1 |
> | MoMo | 0.8 | 0.6 |
> | LDMVFI | 0.9 | 0.5 |
>
> ## Diffusion Distillation
> We do not utilize diffusion distillation techniques because ***we lack a teacher model*** that can robustly interpolate to large, complex motions, as shown in Figures 4 and 5 of our manuscript and the supplementary video demo. All existing video frame interpolation methods cannot interpolate well to large and complex motions. The proposed RDVFI aims to address the limitations of existing video interpolation methods for large, complex motions, rather than approximating the baseline results with fewer sampling steps.
>
> We do not compare with distillation-based interpolation methods due to two main reasons. Firstly, current diffusion-based video frame interpolation methods still focus on addressing their limitations for large, complex motions. To the best of our knowledge, there are ***no existing diffusion distillation baselines*** for video frame interpolation. Secondly, our RDVFI utilizes one-step diffusion sampling because it is only used to determine ambiguous intermediate motions across input frames, rather than fine-grained interpolated frames. The ***motivation and target of the proposed RDVFI is totally different*** from the distillation technique.
>
> ## Number of Keyframes
> We provide an ablation study on keyframe numbers in the FCVG dataset. Because RDVFI-D only interpolates 4N keyframes, we degrade the RDVFI-U due to its flexibility. As we can observe, as the keyframe number increases, the interpolation quality improves accordingly due to its growing ability to model intermediate motions across frames and more accurate optical flow estimation. The gain becomes moderate as the number increases, as more and more motions can be well-modeled and solved with the generated keyframes. Generating more frames with the diffusion model only introduces a marginal performance gain, which supports our claim that generating a few keyframes is sufficient for interpolation and results in superior efficiency.
>
> | Keyframes Number    | LPIPS | FID   | FVD    |
> |---------------------|-------|-------|--------|
> | 1                   | 0.231 | 33.46 | 297.71 |
> | 2                   | 0.227 | 31.19 | 272.31 |
> | 3                   | 0.224 | 29.83 | 257.49 |
> | 4                   | 0.225 | 28.37 | 242.08 |
> | 5                   | 0.224 | 27.22 | 229.91 |
> | 6                   | 0.221 | 27.10 | 220.24 |
> | 7 (Default setting) | 0.220 | 27.03 | 217.72 |
> | 8                   | 0.221 | 27.01 | 218.11 |

---

### Author Response · Authors · 2025-11-28

Dear reviewers:

We appreciate the time you dedicated to reviewing our work. Regarding the concerns you raised, we have provided explanations in our responses. We would like to ensure that your concerns have been adequately addressed. If there are any aspects of our work that remain unclear to you, please don't hesitate to let us know.

Best regards,

Authors of #324

---

### Author Response · Authors · 2025-12-03
**Summary**

Dear Area Chair,



We deeply appreciate your willingness to dedicate your valuable time to conducting a thorough review of our submission and rebuttal document. To assist your evaluation, we provide a brief summary of our work.



**Paper Overview**

Current video diffusion-based methods interpolate as conditional frame generation, thereby requiring tedious diffusion sampling for intermediate frames with high fidelity. This makes these methods extremely slow, usually costing minutes for each interpolation.

To solve this challenge, we propose RDVFI, ***a real-time video frame interpolation method***.
Our RDVFI utilizes the video diffusion model to generate ***continuous pixel trajectories*** rather than frames, which enables one-step diffusion sampling at a relatively small resolution, significantly improving efficiency. We then sample these trajectories at any intermediate timestep for optical flows and warp input frames for interpolation. Our RDVFI requires a video diffusion model that generates accurate motion to warp for interpolation, thus producing advanced interpolation results compared to those of conditional generation baseline methods, which suffer from pixel reconstruction training objectives.

Our RDVFI consistently outperforms existing baseline methods on numeric comparisons across various benchmark datasets. Additionally, most viewers select our RDVFI as the best-performing interpolation method in a subjective evaluation, ***surpassing the second-best method, Wan, for 60% with $50\times$ acceleration.***

**Recognized Strengths by Reviewers**
1. Our RDVFI improves interpolation ***efficiency*** (all reviewers) and ***stablity*** (R4Dp);
2. ***Novel network design*** (all reviewers) that supports ***flexible*** interpolation for various resolutions and interpolation ratio (G3iW);
3. ***Impressive visual improvements*** (except Ba4o).


**Main Concerns Resolved**

We regret that reviewers have not yet responded to our rebuttal. However, we have provided detailed responses and conducted extensive new experiments to fully resolve their initial concerns, which can be summarised as follows:

1. Human evaluation: Reviewer Ba4o claims that they cannot find visual differences between our RDVFI and a baseline method, Wan, which conflicts with the observations of all three other reviewers. To better illustrate our visual improvements, we provide a human evaluation, and our RDVFI receives 60% more selections as the best-performing method than Wan, consistently demonstrating the visual gain of our RDVFI over state-of-the-art methods.
2. More experiments: We provide additional ablation studies and comparisons with other baseline methods. Results clearly show that our RDVFI consistently achieves the best performance, demonstrating the effectiveness of our RDVFI.
3. More explanations and comparisons: We provide more method explanations and comparisons with state-of-the-art methods to illustrate our contributions better.


We sincerely thank all reviewers for their constructive feedback. We believe that our revisions, supplementary experiments, and detailed responses have meticulously addressed all concerns. We kindly request that our rebuttal be fully considered in your final decision. We are truly grateful for your time and expertise throughout this review process.

Best regards,

Authors of #324

---

### Meta-Review · Area_Chair_erX3 · 2026-01-11

**Summary:**

The paper proposes a novel technique for video frame interpolation, that decouples the process into two stages of motion estimation and appearance generation, mixing with one-step diffusion, and showing improved efficiency and quality compared to select baselines.

Reviewers expressed concerns about breadth of evaluations, claims of quality improvements, limited test sets, missing baselines and related work, but most of these were carefully addressed in the rebuttal.

**Reviewer Concerns:**

Reviewers expressed concerns about:

- Missing efficiency metrics (addressed)
- Ablations on number of keyframes (addressed)
- Limited test set (addressed)
- Quality (perhaps partially addressed with a user study, although it's not clear whether this would convince the reviewer, as it's a subjective claim)
- Failure cases (addressed)
- Baselines and related work (addressed)

Remaining concerns that were not addressed:

- A direct distillation baseline (Ba4o)
- Replacing the VAE encoder with something like DINOv2 (G3iW)

^ In this AC's view, neither of these two seem significant.

**Reviewer Scores:**

Two reviewers rated the paper as borderline reject. One reviewer rated as borderline accept. One reviewer rated as reject.

I believe after the rebuttal, the two borderline rejects would have increased to borderline accepts. I don't think the reject would have changed score. If anything, perhaps to borderline reject.

So, the final scores would likely have been (6, 6, 6, 2) or (6, 6, 6, 3).

---

### Decision · Program_Chairs · 2026-01-26

Accept (Poster)